# A Liposomal Formulation Enhances the Anti-Senescence Properties of Nicotinamide Adenine-Dinucleotide (NAD^+^) in Endothelial Cells and Keratinocytes

**DOI:** 10.3390/cimb47090722

**Published:** 2025-09-05

**Authors:** Stefano Ministrini, Luca Liberale, Hanns-Eberhard Erle, Giuseppe Percoco, Ali Tfayli, Ali Assi, Ivan Kapitonov, Isabel Greiner, Giovanni Guido Camici

**Affiliations:** 1Center for Molecular Cardiology, Schlieren Campus, University of Zurich, Wagistrasse 12, 8952 Schlieren, Switzerland; stefano.ministrini@uzh.ch; 2First Clinic of Internal Medicine, Department of Internal Medicine, University of Genoa, 6 Viale Benedetto XV, 16132 Genoa, Italy; luca.liberale@unige.it; 3IRCCS Ospedale Policlinico San Martino Genoa—Italian Cardiovascular Network, L.go R. Benzi 10, 16132 Genoa, Italy; 4Independent Researcher, Chemin des Macherettes 28, 1172 Bougy Villars, Switzerland; hanns_erle@yahoo.com; 5Eurofins BIO-EC Laboratory, Chemin de Saulxier 1, 91160 Longjumeau, France; giuseppe.percoco@cpt.eurofinseu.com; 6Unité Universitaire Interdisciplinaire Lip (Sys), Lipides, Systèmes Analytiques et Biologiques, Faculté de Pharmacie, Université Paris-Saclay, 17 Avenue des Sciences, 91400 Orsay, France; ali.tfayli@upsud.fr (A.T.); ali.assi@universite-paris-saclay.fr (A.A.); 7Sovida Solutions Limited, Michelin House, 81 Fulham Road, London SW3 6RD, UK; ik@ivkapmail.com (I.K.); isabel@sovida.com (I.G.); 8Department of Research and Education, University Hospital Zurich, Rämistrasse 100, 8092 Zurich, Switzerland

**Keywords:** nicotinamide adenine-dinucleotide, liposomes, microcirculation, keratinocytes, anti-senescence, molecular mechanisms

## Abstract

Nicotinamide adenine-dinucleotide (NAD^+^) supplementation is a promising strategy to delay cellular aging in different areas, including cosmetic dermatology. However, low bioavailability and stability of NAD^+^ formulations are the main factors limiting its effectiveness as an anti-aging treatment. In light of the above, a liposomal formulation of NAD^+^ (LF-NAD^+^) was tested in this study and compared to NAD^+^ alone in primary human aortic endothelial cells (HAECs) and primary human epidermal keratinocytes (HEKas). Intracellular NAD^+^ was measured using a colorimetric assay. Cell survival was derived from lactate dehydrogenase release in supernatants. Cell senescence was measured by senescence-associated β-galactosidase staining. Molecular mechanisms underlying the reported effects were analyzed by Western blot. Skin penetration of NAD^+^ was measured ex vivo in skin explants, using infrared spectroscopy. Compared to control NAD^+^ alone, the LF-NAD^+^ formulation increased the intracellular NAD^+^ content and cell survival in HAECs, but not in HEKas. Instead, a significant reduction in the number of senescent cells was observed in both HAECs and HEKas. LF-NAD^+^ treatment was associated with a reduced expression of *p16* in both HAECs and HEKas, and to a significant reduction in p21 in HEKas alone. Finally, LF-NAD^+^ increases the skin penetration of the active substance NAD^+^ by 30% compared to the application of NAD^+^ alone. LF-NAD^+^, enhances the anti-aging effects of NAD^+^ on vascular and skin cells. Such in vitro findings might indicate a potential anti-aging role in the microcirculation and in the epidermidis.

## 1. Introduction

Nicotinamide adenine dinucleotide (NAD^+^) is a co-enzyme involved in multiple cellular functions and crucial for cellular metabolism. The reduced form of NAD^+^ (NADH) is a metabolite of glycolysis and plays a pivotal role in cellular energy metabolism and redox balance. Furthermore, many enzymes employ NAD^+^ as a substrate or co-factor, generating its metabolite nicotinamide (NAM) as a by-product. NAD^+^-consuming enzymes belong to three classes: NAD^+^ glycohydrolases, sirtuins and poly (ADP-ribose) polymerases (PARPs) [1]. All these enzymes are involved in cellular senescence, tissue aging and longevity: specifically, the activation of sirtuins is associated with delayed senescence and prolonged lifespan in mammals, whereas NAD^+^ glycohydrolases and PARPs have opposite effects [2,3,4].

In physiologic conditions, the intracellular pools of NAD^+^ and NADH are in a dynamic balance, depending on the redox state of the cell and its mitochondrial efficiency, with NAD^+^ being the most abundant form. Indeed, NADH is generated by the glycolysis and Krebs cycle, then quickly oxidized to NAD^+^ in the mitochondrial electron transport chain [5].

In order to maintain a constant intracellular pool, NAD^+^ can be generated from its precursor NAM or synthesized de novo in the liver. Interestingly, aging is characterized by a gradual depletion of NAD^+^ intracellular pool, due to an imbalance between catabolic and anabolic processes. For this reason, NAD^+^ supplementation has been proposed as a therapeutic strategy for aging-related disorders [1]. In particular, in the field of cosmetic dermatology, it is a well-established topical treatment to delay skin aging and improve skin tone [6,7]. Nonetheless, bioavailability and stability of NAD^+^ are the main factors limiting its effectiveness as topical treatment. Indeed, extracellular NAD^+^ is rapidly metabolized to adenosine and then internalized by skin fibroblasts [8].

Keratinocytes and endothelial cells play a pivotal role in skin aging. Epidermidis constitutes the external layer of the skin and exerts a protective function against physical and chemical external agents. Integrity of epidermidis relies on the constant turn-over of keratinocytes, the most abundant cell type present in the epidermis. Keratinocytes replicate in the basal layer and then maturate into keratin-producing cells, lose their nucleus and build the outer keratin layer, where they progressively exfoliate [9]. Endothelial cells are the most abundant cells in microcirculation and play a fundamental role in maintaining skin trophism and tone. Furthermore, the skin’s microcirculation contributes to thermoregulation, immune response and wound healing [10]. Aging skin, like every other aging organ, is characterized by the accumulation of senescent cells. In turn, senescent cells are characterized by the exhaustion of their proliferative capacity, alongside the acquisition of the senescence-associated secretive phenotype (SASP), which promotes inflammation and induces senescence in neighboring cells with a paracrine mechanism, thus contributing to aging-associated changes in the skin [11,12,13]. Senescence of keratinocytes in the basal layer impairs the proliferative capacity of epidermidis and, thus, its physiologic turn-over. This causes the thinning of the epidermidis, and the reduction in the contact surface area between the dermis and epidermis in the dermal–epidermal junction, where the outer microvascular plexus lies, leading to a reduced oxygen and nutrition supply to the epidermis and, consequently, a further impairment of basal cells proliferation [14]. Indeed, aging is associated with a reduction in blood flow to the skin, with a 40% sink between the ages of 20 and 70 [15], which reflects anatomical changes in the microcirculation. These changes can be secondary to SASP in keratinocytes but also to the primary senescence of endothelial cells [16]. The reduction in NAD^+^ intracellular pool is a strong drive for endothelial senescence and microvascular aging, as it impairs the function of pivotal NAD-dependent enzymes, such as Sirtuins and PARPs, regulating endothelial survival, vascular reactivity and inflammation [3,17,18]. Besides being an esthetic issue, skin aging also increases the risk of chronic wounds, infections, inflammatory skin diseases (e.g., bullous pemphigoid) and malignancies [19]. Finally, senescent dermal fibroblasts contribute to skin aging, resulting in a reduced production of collagen synthesis, an increased secretion of disintegrin and matrix metalloproteinases, and weakening of the dermoepidermal junction [20]. However, the role of NAD^+^ and NAD^+^ supplementation in dermal fibroblasts was previously investigated [21].

*p21* and *p16* are commonly employed as biomarkers of cellular senescence [22,23]. *p16* and *p21* inhibit the binding between the cyclin dependent kinases (CDK) 2, 4 and 6 and their target cyclins, namely cyclin D and cyclin E. When bound to cyclins, CDKs phosphorylate the protein Rb, promoting the progression of the cell cycle and, eventually, cellular replication. By preventing the formation of the CDK–cyclin complex, *p16* and *p21* induce the arrest of the cell cycle and the transition to cellular senescence [24,25,26].

Liposome encapsulation is a pharmaceutical technology employed to increase the stability of various compounds and to enhance the delivery of a hydrophilic drug beyond the cellular membrane barrier. This technology has been successfully employed for anti-neoplastic drugs and analgesics [27]. Liposomes consist of one or more lipid bilayers, containing phospholipids and some stabilizers, such as cholesterol. Lipophilic molecules can be encapsulated in the hydrocarbon interface between the two phospholipid layers, making them soluble in an aqueous medium, whereas hydrophilic molecules can be enclosed in the inner part of the liposomes, protecting them from the interaction with other soluble substances like oxidizing agents or lytic enzymes [28]. We hypothesized that liposome encapsulation may improve the bioavailability and stability of NAD^+^, thus increasing its effectiveness as an anti-aging treatment. To this end we studied primary human endothelial cells whose function is crucial for delivering oxygen and nutrients to the skin, and keratinocytes, the most abundant cell type in the epidermis. Finally, we tested skin penetration of encapsulated NAD^+^. To the best of our knowledge, this is the first time that a liposomal formulation of NAD^+^ has been tested in the setting of anti-aging cosmetology.

The aim of this study was to investigate whether a liposomal formulation of NAD^+^ (LF-NAD^+^) is superior to NAD^+^ alone in penetrating skin, preserving cell survival and reducing cell senescence of primary human endothelial and epidermal cells.

## 2. Materials and Methods

### 2.1. Liposomal Formulation

The liposome matrix was kindly provided by Sovida Solutions Ltd., London, UK (patent nr. US 12,150,461 B2). The composition of the liposome matrix is reported in Table 1. All the components of the liposomal formulation are listed within the International Nomenclature of Cosmetic Ingredients (INCI). Chemical and physical features of the liposomal matrix and LF-NAD+ are reported in Appendix A. Size and over-time stability of liposomes where characterized by laser diffraction [29], as shown in Appendix A.

### 2.2. Cell Culture

Primary human aortic endothelial cells (HAECs—Lonza Bioscience, Basel, Switzerland) were purchased at passage 2 and expanded until passage 7 in endothelial growth medium 2 (EGM-2—Lonza Bioscience, Basel, Switzerland) supplemented with 10% fetal bovine serum (FBS). Cells were detached by Tripsin/EDTA and reseeded in 6-well plates ($1.8\times 10^{5}$/well). Cells were grown to 80% confluence and rendered quiescent for 24 h in medium containing 0.5% FBS [30]. Next, cells were treated with either NAD^+^ 200 μM or NAD^+^ 200 μM in LF-NAD^+^ diluted in EGM-2 + 0.5% FBS. The ultimate vol/vol concentration of LF-NAD^+^ in EGM-2 + 0.5% FBS is 0.225%. Control condition was EGM-2 + 0.5% FBS. Primary human epidermal keratinocytes adult (HEKas—Lonza Bioscience, Basel, Switzerland) were purchased at passage 2 and expanded to passage 3 in dermal basal cell medium (DCBM, American Type Culture Collection, Manassas, VA, USA). Cells were detached by using Tripsin/EDTA and reseeded in 6-well plates ($1.8\times 10^{5}$/well) [31]. Cells were grown to 60% confluence and treated with either NAD^+^ 200 μM or LF-NAD^+^, as described above. Control condition was DCBM.

### 2.3. Intracellular NAD^+^/NADH Measurement

Intracellular content of NAD^+^ was measured using a colorimetric assay (ab221821, Abcam, Cambridge, UK) [32]. Cells were cultivated as described above and treated with either NAD^+^ 200 μM, LF-NAD^+^ or control for 90 min, then detached by using Tripsin/EDTA and lysed according to the manufacturer’s instructions to extract NAD^+^ and NADH. Concentrations of NAD^+^ and NADH were then measured using a microplate absorbance reader (Tecan, Männendorf, Switzerland). Protein concentration in cell lysates was determined by Bradford reaction, according to the manufacturer’s recommendations (Bio-Rad Laboratories AG, Fribourg, Switzerland). Concentrations of NAD^+^ and NADH were then normalized to intracellular protein content.

### 2.4. Cell Survival

Lactate dehydrogenase (LDH) release was measured in the supernatants of cells treated with either NAD^+^ 200 μM, LF-NAD^+^ or control for 48 h using a colorimetric assay (Roche Diagnostics GmBH, Mannheim, Germany). Absorbance was measured using a 490 nm wavelength (reference > 600 nm) in a microplate reader (Tecan, Männendorf, Switzerland). After subtraction of blank absorbance, the absorbance of each sample was subtracted to the absorbance of the positive control and then normalized NAD^+^ 200 μM, to obtain the cell survival (see formula below). Experimental positive control was achieved by treatment with Triton-X100 0.01%.
(1)$$Cell\;survival\;\left(\%\right)=\frac{{Abs}_{positive\;control}\left(OD\right)-{Abs}_{sample}(OD)}{{Abs}_{positive\;control}\left(OD\right)-{Abs}_{negative\;control}(OD)}\times 100$$

### 2.5. Cell Senescence

Cells were expanded as described above, then detached by using Tripsin/EDTA and reseeded in 2-well slides (10^5^/well). Thereafter, cells were treated with NAD^+^ 200 μM or LF-NAD^+^ for 48 h. Medium was changed every 24 h. Senescent cells were assessed by staining for senescence-associated β-galactosidase (SABG) following the indications of the manufacturer (Merck KGaA, Darmstadt, Germany) [33]. The percentage of positively stained cells was calculated by averaging counts from four visual fields for each slide using a microscope (magnification factor 10x) and normalized to NAD^+^ 200 μM. All pictures have been standardized in terms of image brightness, contrast and resolution before the analysis.

### 2.6. Western Blot

Cells were treated as described in Section 2.2. After removal of supernatants, cells lysed using a buffer containing Tris 50 mM, NaCl 150 mM, EDTA 1 mM, NaF 1 mM, DTT 1 mM, aprotinin 10 mg/mL, leupeptin 10 mg/mL, Na_3_VO_4_ 0.1 mM, phenylmethylsulfonyl fluoride (PMSF) 1 mM and NP-40 0.5%. Protein concentration was determined as described above. An amount of 20 µg of total protein lysates was separated on a 10% SDS–PAGE before being transferred to a polyvinylidene fluoride membrane using a wet transfer method (Bio-Rad). Membranes were incubated with primary antibodies against *p21* (1:2000, Santa Cruz Biotechnology, Santa Cruz, CA, USA) and *p16* (1:1000, Santa Cruz Biotechnology, Santa Cruz, CA, USA) at 4 °C overnight on a shaker. The following incubation with secondary antibody (anti-mouse 1:2000, Southern Biotechnology, Birmingham, AL, USA) was performed for 1 h at room temperature. Densitometric analyses were performed (Amersham Imager 600, GE Healthcare Europe GmbH, Glattbrugg, Switzerland), and protein expression was normalized to GAPDH.

### 2.7. Ex Vivo Skin Penetration

Circular skin explants (Ø = 38 mm) were prepared from the abdominal skin from a Caucasian, 29-year-old, anonymous donor. The collection process fully complied with Articles L.1245-2 and L.1211-1 of the French Public Health Code. The donor provided written informed consent prior to tissue procurement. The explants were prepared as previously described [34]. Briefly, skin samples were held on a specifically designed support composed of a reservoir of culture medium surmounted by a grid on which the skin is stretched. The skin support was connected by a fluidic circuit to a second reservoir of culture medium which was stored in the incubator at 37 °C, 5% CO_2_. The circulation of the culture medium was ensured by a peristaltic pump. Each skin explant was treated with 9 µL (2 µL/cm^2^) of NAD^+^ 200 μM or LF-NAD^+^ applied with a small spatula. The control explants did not receive any treatment. After a 24 h treatment, samples were fixed in buffered formalin solution and included in paraffin for histological sections. 10 µm thick sections were made using a rotatory microtome (Leica RM 2125, Wetzlar, Germany), and the sections were mounted on a CaF2 specific support for infrared spectroscopy imaging analysis. Based on a preliminary infrared spectral analysis of the pure compounds, NAD^+^ shows a peak of absorbance at a wavenumber of 1035 cm^−1^ (Appendix A). This peak was used to determine the penetration of NAD^+^ into the skin. Data collection was performed with a Spotlight 400 FT-IR imaging system (PerkinElmer Inc., Shelton, CT, USA), over a spectrum of 750–4000 cm^−1^. Data were collected over a surface of 0.5 mm^2^ with a spatial resolution (pixel size) of 6.25 × 6.25 µm^2^. Hyperspectral images were reconstructed, associating each pixel to a color, representing the intensity of absorbance in that point. Red color indicates a higher intensity, whereas blue indicates a low intensity. The concentration of NAD^+^ inside of the skin was calculated using the non-negativity constraint classical least squares (NCLS) model. NCLS is a spectral unmixing method that aims to estimate the concentrations of known spectral signatures in a spectrum, as previously described [35], to estimate the abundance fractions of a known spectral signature as a linear combination of each component of the signature. The NCLS score is calculated per each pixel and quantified as mean ± SEM.

### 2.8. Statistical Analysis

Values are expressed as mean ± SEM. A significance threshold for type I probability of error was set <0.05. Comparisons between two groups were performed using the unpaired t Test. Comparisons among multiple groups were performed using analysis of variance (ANOVA) test with appropriate post hoc correction for multiple comparisons. Statistical analysis was conducted using GraphPad Prism 8 (GraphPad Software Inc. Boston, MA, USA).

## 3. Results

### 3.1. Intracellular Delivery of NAD^+^ and NADH

First, we aimed at investigating whether the LF-NAD^+^ formulation increases the intracellular delivery of NAD^+^ compared to NAD^+^ alone and negative control. A significant increase in intracellular NAD^+^ content was observed in HAECs after LF-NAD^+^ treatment, but not with NAD^+^ 200 μM (Figure 1A). No significant difference among the three conditions was observed in NADH content and in NADH/NAD^+^ ratio (Figure 1B,C). Conversely, in HEKAs the intracellular NAD^+^ content was reduced compared to negative control (Figure 1D), suggesting that NAD^+^ is quickly converted to NADH, as confirmed by a significantly increased NADH/NAD^+^ ratio (Figure 1E,F). These results suggest that the LF-NAD^+^ formulation increases the intracellular NADH/NAD^+^ pool within 90 min from administration, though with some differences between the cell types.

### 3.2. Effect of LF-NAD^+^ on Cell Survival and Senescence

Then, we investigated the functional effects of LF-NAD^+^ formulation, in terms of cell survival and senescence. A significant increase in survival was observed in HAECs treated with LF-NAD^+^, compared to control +19.3% (95% C.I. 18.4–20.1%) and to NAD^+^ alone +4.3% (95% C.I. 3.8–4.7%) (Figure 2A). No significant difference was instead observed in HEKas: −0.5% (95% C.I. −24.2–25.1%) compared to negative control and +8.2% (95% C.I. −2.9–18.7%) compared to NAD^+^ 200 μM (Figure 2B).

A significant reduction in the number of senescent cells was observed in both HAECs and in HEKas treated with LF-NAD^+^. Compared to NAD^+^ alone, the average reduction was 28.7% (95% C.I. 9.4–33.9%) and 15.4% (95% C.I. 4.9–24.3%) for HAECs and HEKas, respectively (Figure 3A–C). Figure 3B,D are representative pictures of SABG staining in cells treated with NAD^+^ alone or LF-NAD^+^.

### 3.3. Effect of LF-NAD^+^ on Molecular Markers of Cell Senescence

To investigate the molecular mechanisms underlying the reduced senescence observed after the administration of LF-NAD^+^, molecular effectors of senescence p16 and p21 were measured. Compared to NAD^+^ alone, the treatment with LF-NAD^+^ was associated with a reduced expression of *p16* in both HAECs and HEKas (Figure 4A–C) and with a significant reduction in *p21* in HEKas only (Figure 4B–D).

### 3.4. Ex Vivo Skin Penetration of NAD^+^

According to the above-reported results, LF-NAD^+^ has an anti-senescence effect on endothelial cells, the main components of the skin microcirculation. Skin microcirculation is organized in two plexuses, parallel to the skin’s surface, in order to supply oxygen and nutrients to both the outer layer (epidermidis) and the inner layer (dermis) of the skin [36]. So, in order to exert its beneficial effect on microcirculation, LF-NAD^+^ should penetrate the skin until the dermoepidermal junction. Using infrared spectroscopy on human skin explants treated with LF-NAD^+^ or NAD^+^ alone, we observed that LF-NAD^+^ increases the skin penetration of the active substance NAD^+^ by 30% compared to the application of NAD^+^ alone (Figure 5).

## 4. Discussion

In the present study, we wanted to test whether a liposomal formulation of NAD^+^ could be superior to NAD^+^ alone in preventing skin aging. Since the main limitation to the use of NAD^+^ in the field of cosmetic dermatology is the limited bioavailability, we wanted to test whether LF-NAD^+^ could increase the delivery of NAD^+^ inside the cells and across the skin layers. Then, we wanted to investigate if the increased NAD^+^ delivery was associated with a functional effect in terms of cell survival and senescence of keratinocytes and endothelial cells. The liposomal formulation was adapted from previously established formulations (e.g., Natipide^®^II) that were previously tested for cutaneous applications [37,38,39].

The results reported in this study demonstrate that LF-NAD^+^, a lipid formulation designed to enhance stability and bioavailability of NAD^+^, is superior to NAD^+^ alone in delivering NAD^+^ inside primary human endothelial and primary human epidermal cells. However, in HEKas the increased NAD^+^ delivery is also associated with an unexpected change in the cell redox balance, as shown by the increase in the NADH/NAD+ ratio. Indeed, in physiologic conditions the NADH/NAD^+^ pool is maintained with a surplus of oxidized NAD^+^, whereas an excess of NADH over NAD^+^ generates a non-physiologic condition named “reductive stress” [40]. The reductive couple NADH/NAD^+^ has a pivotal role in binding the cell redox state with energy production; indeed, NAD^+^ is employed in three different steps of the Krebs’ cycle, whereas NADH provides electrons to the mitochondrial electrons transport chain, and eventually to oxidative phosphorylation. Furthermore, β-oxidation of fatty acids requires NAD^+^ as an oxidizing factor, whist the biosynthesis of proline from glutamate involves NADH as a co-factor [41]. So, reductive stress affects multiple steps of cell bioenergetics. However, whilst the role of oxidative stress in cellular senescence and aging is well-known, the role of reductive stress is still largely unclear.

We also observed that the treatment with LF-NAD^+^ reduces cellular senescence of both HAECS and HEKas, compared to NAD^+^ alone. At the same time, LF-NAD^+^ is superior to NAD^+^ alone in promoting cell survival of primary human endothelial cells. These effects can all be attributed to the increased intracellular delivery of NAD^+^ in HAECs. Furthermore, the derangement in the redox balance of HEKAs, induced by LF-NAD^+^ treatment, could explain the neutral effect of LF-NAD^+^ on survival of this cell type. Indeed, recent in vitro evidence in breast cancer cell lines suggests that reductive stress has a more pronounced cytotoxic effect on senescent cells [42]. According to this hypothesis, LF-NAD^+^ could have a senolytic effect in HEKas by preferentially inducing cell death in senescent cells. Accordingly, the neutral effect on cell survival would be the result of the preferential induction of cell death in senescent cells and the promotion of survival in non-senescent cells, like the one observed in HAECs. Accumulation of senescent cells exerts a negative effect in adult tissues, as described above. Removal of senescent cells by inducing their apoptosis is therefore considered a potential anti-aging intervention [43].

Compared to NAD^+^ alone, the LF-NAD^+^ formulation reduced the expression of two molecular effectors of cell senescence, namely *p16* and *p21* [44,45], providing a potential mechanistic explanation for its higher effectiveness. Indeed, *p21* and *p16* are commonly employed as biomarkers of cellular senescence [22,23], in addition to SABG staining, reinforcing the evidence of an increased anti-senescence effect of LF-NAD^+^. The main transcriptional regulator of *p21* is *p53*, which is in turn de-acetylated by the NAD+-dependent enzymes sirtuins 1, 2, 6 and 7 [46]. Since de-acetylation of *p53* is associated with a reduced transcriptional activity [47], we hypothesize that the liposomal formulation of NAD+ delivers a higher amount of NAD+ inside the cells, thus yielding a stronger inhibition of *p21*. Conversely, *p16* is not regulated by *p53* and it constitutes, according to previous reports, an alternative pathway of cell senescence to the *p53*/*p21* axis [48,49]. The regulation of *p16* is a complex system, encompassing multiple genetic and epigenetic mechanisms [49]. Interestingly, *p16* can be independently activated by reactive oxygen species (ROS) [48], originating from the mitochondria and scavenged by the mitochondrial sirtuins 3, 4 and 5 [50]. Accordingly, we hypothesize that the increased delivery of NAD^+^ by liposomal encapsulation may increase the activity of the mitochondrial sirtuins, and that the reduced *p16* expression reflects the reduced amount of intracellular ROS. According to the presented data, this latter mechanism seems to be prevalent in endothelial cells, whereas both pathways are activated in keratinocytes.

From a translational perspective and building on the herein reported results, treatment with LF-NAD^+^ could yield superior anti-aging effects compared to NAD^+^ alone in the setting of cosmetic dermatology. In particular, the results on endothelial cells suggest that LF-NAD^+^ could preserve the viability and functionality of the skin microcirculation, which is a crucial factor for skin trophism and tone [10]. According to this hypothesis, the liposomal formulation could be employed in topical anti-aging cosmetic products, such as creams or sera. Compared to existing NAD^+^-based products, the use of LF-NAD^+^ would provide the additional benefit of increasing the intracellular delivery of NAD^+^, especially in the microcirculatory plexuses of the skin. As previously described, these plexuses are located in the deepest layers of skin, so an optimal skin penetrance is crucial to allow the effect of LF-NAD+ on microcirculation. To this aim, the results of ex vivo experiments in human skin explants confirm that LF-NAD+ has an increased skin penetrance compared to NAD+ alone. To the best of our knowledge, this is the first time that a liposomal formulation of NAD+ is tested ex vivo and in vitro. However, additional investigations, such as clinical trials, are needed to confirm that improvements in skin tone and texture can be significantly appreciated by potential users.

Some limitations must be acknowledged. Firstly, employed cell cultures were not in a condition of advanced senescence, as can be appreciated in the representative figures of SABG staining. This choice was dictated by the accelerated process of senescence observed in cultivated primary keratinocytes, which reach a senescence plateau after few passages [51]. Despite the use of primary human cells, 2D cell cultures do not account for the complex interaction of skin fibroblasts, keratinocytes and endothelial cells. Similarly, they do not account for variability of human skin across age, sex and ethnicity. Additionally, the proposed mechanistic insights are associative and would require additional rescue experiments. So, results of this in vitro study warrant further research in vivo to confirm the superior anti-aging properties of LF-NAD^+^ compared to NAD^+^ alone. Furthermore, the endothelial cells used were of macrovascular origin, rather than of a micro-vascular one. This point deserves mentioning since the differences between HAECs and dermal microvascular have not been investigated previously. Nonetheless, HAECs (as well as umbilical vein endothelial cells) are usually considered a reliable universal model to study endothelial biology. Finally, the stability of liposome matrix in different storage conditions, the potential aggregation of liposome, and the effects on ultraviolet-light-induced senescence have not been addressed in this paper and should be assessed in future studies.

## 5. Conclusions

The present study shows that LF-NAD^+^ maximizes the anti-senescence effects of NAD^+^ in endothelial cells and keratinocytes. This effect is likely due to an increased skin penetrance and a higher intracellular availability of NAD^+^. Although in vivo confirmation is needed, the herein reported findings suggest a potentially superior anti-aging effect of LF-NAD^+^ compared to NAD^+^ alone, with potential implications in the field of cosmetic dermatology.

## Figures and Tables

**Figure 1 cimb-47-00722-f001:**
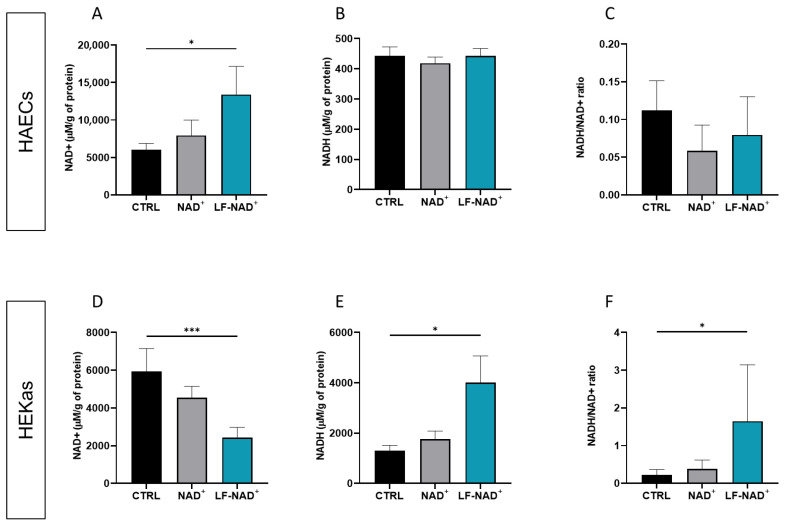
**Effects** **of the liposomal formulation of oxidized nicotinamide adenine-dinucleotide (NAD^+^) on the intracellular oxidized/reduced NAD^+^/NADH pool.** Intracellular contents of NAD^+^ (**A**), NADH (**B**) and NAD^+^/NADH ratio (**C**) were measured in human aortic endothelial cells (HAECs) after a 90 min treatment with either medium control (CTRL), NAD+ alone or the liposomal formulation of NAD^+^ (LF-NAD^+^). Intracellular contents of NAD^+^ (**D**), NADH (**E**) and NAD^+^/NADH ratio (**F**) were measured in human epidermal keratinocytes adult (HEKas) after a 90 min treatment with either medium control, NAD^+^ alone or LF-NAD^+^. One-way repeated measures analysis of variance (ANOVA) with Sidak’s post hoc correction, n = 6; * *p* < 0.05, *** *p* < 0.001.

**Figure 2 cimb-47-00722-f002:**
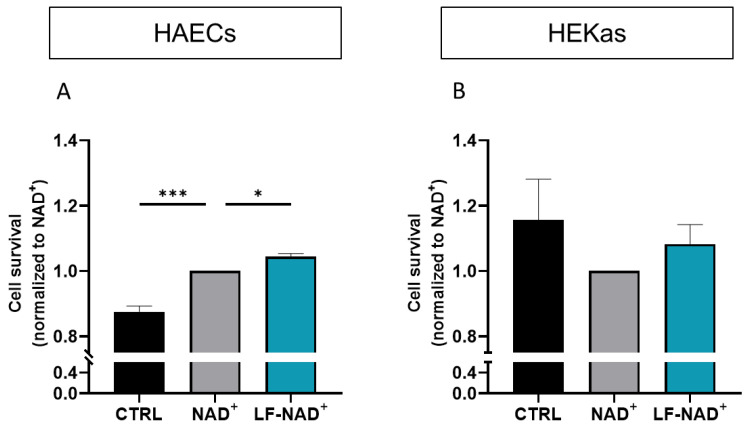
**Effects of the liposomal formulation of oxidized nicotinamide adenine-dinucleotide (NAD^+^) on cell survival.** Death rate of human aortic endothelial cells (HAECs) was measured as lactate dehydrogenase (LDH) assay after a 48 h treatment with either medium control (CTRL), NAD^+^ alone or the liposomal formulation of NAD^+^ (LF-NAD^+^). (**A**). Death rate of human epidermal keratinocytes adult (HEKas) was measured as lactate dehydrogenase (LDH) assay after a 48 h treatment with either medium control (CTRL), NAD^+^ alone or LF-NAD^+^ (**B**). Values are normalized to NAD^+^ alone (ratio). One-way analysis of variance (ANOVA) with Tukey’s post hoc correction, n = 6; * *p* < 0.05, *** *p* < 0.001.

**Figure 3 cimb-47-00722-f003:**
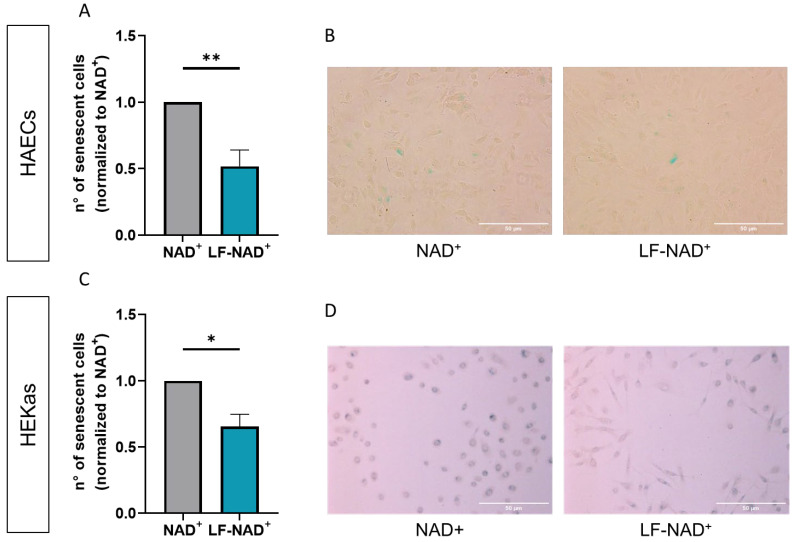
**Effects of the liposomal formulation of oxidized nicotinamide adenine-dinucleotide (NAD^+^) on cell senescence.** Cell senescence was measured by senescence-associated β-galactosidase (SABG) staining in human aortic endothelial cells (HAECs) after a 48 h treatment with NAD^+^ alone or the liposomal formulation of NAD^+^ (LF-NAD^+^) for 48 h (**A**). Representative pictures of SABG staining in HAECs (**B**). Cell senescence was measured by SABG in human epidermal keratinocytes adult (HEKas) after a 48 h treatment with NAD^+^ alone or LF-NAD^+^ (**C**). Representative pictures of SABG staining in HEKas. Magnification factor 10× (**D**). Values are normalized to NAD^+^ alone (ratio). Student’s t test, n = 3; * *p* < 0.05, ** *p* < 0.01.

**Figure 4 cimb-47-00722-f004:**
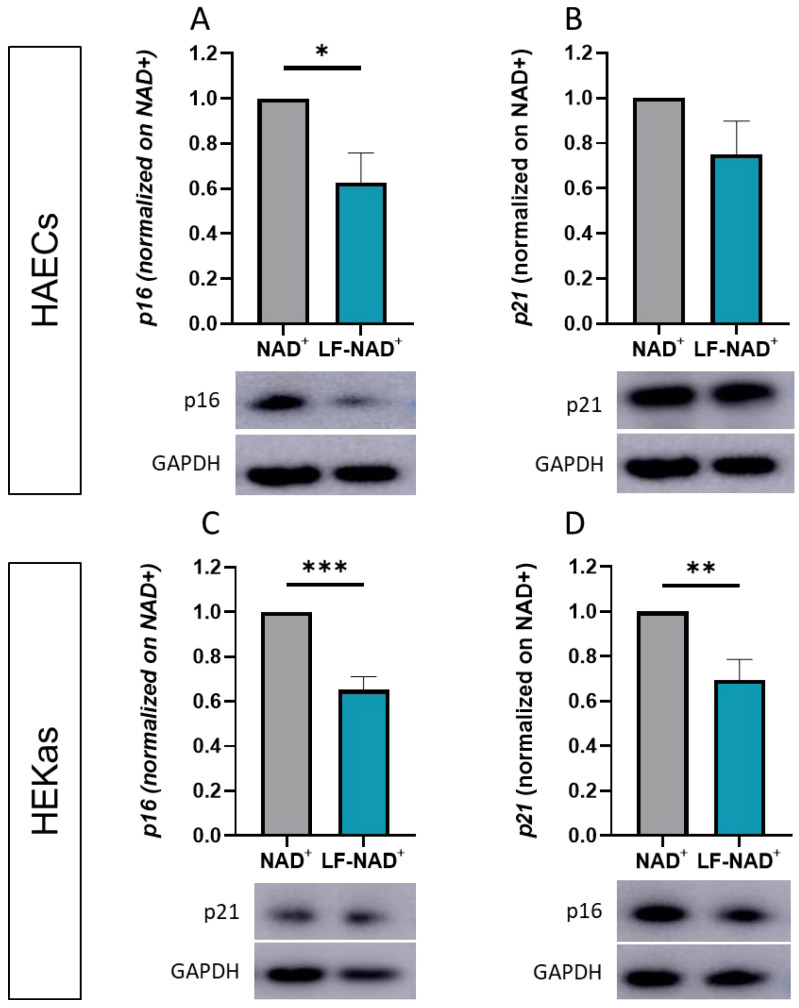
**Effects of the liposomal formulation of oxidized nicotinamide adenine-dinucleotide (NAD^+^) on the expression of** ***p16*** **and** ***p21*****.** Expression of senescence-associated proteins *p16* (**A**) and *p21* (**B**) was measured by Western blot in human aortic endothelial cells (HAECs) after a 48 h treatment with NAD^+^ alone or liposomal formulation of NAD^+^ (LF-NAD^+^). Expression of senescence-associated proteins *p16* (**C**) and *p21* (**D**) in human epidermal keratinocytes adult (HEKas) after a 48 h treatment with NAD^+^ alone or LF-NAD^+^. Glyceraldehyde 3-phosphate dehydrogenase (GAPDH) was used as loading control. Values are normalized to NAD^+^ alone (ratio). Student’s t test, n = 6; * *p* < 0.05, ** *p* < 0.01, *** *p* < 0.001.

**Figure 5 cimb-47-00722-f005:**
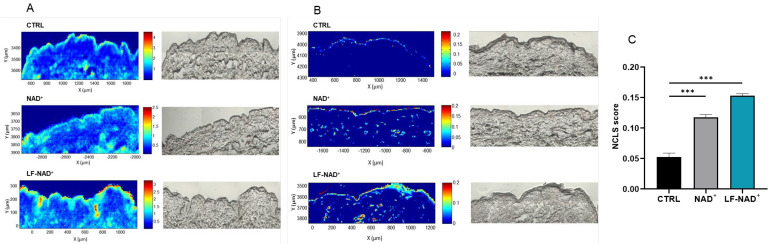
**Skin penetration of oxidized nicotinamide adenine-dinucleotide (NAD^+^) in the skin, assessed by infrared spectroscopy.** Infrared absorbance band at 1035 cm^−1^ wavenumber in untreated skin (CTRL), skin treated with NAD^+^ alone or liposomal formulation of NAD^+^ (LF-NAD^+^) (**A**). Quantification NAD^+^ content in untreated skin (CTRL), skin treated with NAD^+^ alone or LF-NAD^+^ (**B**). Average NAD^+^ content in untreated skin (CTRL), skin treated with NAD^+^ alone or LF-NAD^+^ (**C**). Values are expressed as non-negativity constraint classical least squares model score (absolute number). One-way analysis of variance (ANOVA) with Tukey’s post hoc correction, n = 1; *** *p* < 0.001.

**Table 1 cimb-47-00722-t001:** Composition of the liposome matrix as provided by Sovida Solutions Ltd. This solution was then diluted 1:3 in distilled water and nicotinamide adenine dinucleotide (NAD^+^) was added at a concentration of 5% *w*/*w*.

Components	Quantity (%)
Pentylene Glycol	4.95
Lecithin	4.21
Sodium hydroxide	0.32 (as 10% solution to adjust pH)
Tocopherol	0.02
Water	24.10
Glycerin	66.40 (as a preservative)

## Data Availability

Data are available upon request to the corresponding author.

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
