# Peer review of "A Liposomal Formulation Enhances the Anti-Senescence Properties of Nicotinamide Adenine-Dinucleotide (NAD^+^) in Endothelial Cells and Keratinocytes"

_cimb, 2025, doi:10.3390/cimb47090722_

Round 1

Reviewer 1 Report

Comments and Suggestions for Authors

The manuscript “A novel liposomal formulation enhances the anti-senescence properties of nicotinamide adenine-dinucleotide (NAD+) in endothelial cells and keratinocytes” is a research article about LIPONAD, a novel liposomal NAD+ formulation, that enhances NAD+ bioavailability and stability reducing senescence in both vascular and skin cells.

The study is well-designed, the findings are solid and the translational applications are interesting. The English language is fine and does not require revision. I suggest the following improvements of the manuscript:

  1. Line 93 and everywhere else in the text: please use the scientific notation to express big numbers (180,000 should be 1,8 * 10^5)
  2. Why NAD+ treated group does not show standard deviation in some figures?
  3. In the legend of figure 4 there is a mistake “figure 1”. Twice written “of”.
  4. Why did authors utilize exactly these cell lines for this study? HAECs are aortic endothelial cells.
  5. The quality of images in figure 3 (B and D) should be improved. Inserting a scale bar or specifying magnification would help the readers.
  6. In the figures NAD+ is sometimes written without uppercase, in the text with. Please reconcile writing it always with uppercase.
  7. Was the liposome matrix tested for toxicity?

Author Response

Reviewer #1

The manuscript “A novel liposomal formulation enhances the anti-senescence properties of nicotinamide adenine-dinucleotide (NAD+) in endothelial cells and keratinocytes” is a research article about LIPONAD, a novel liposomal NAD+ formulation, that enhances NAD+ bioavailability and stability reducing senescence in both vascular and skin cells.

The study is well-designed, the findings are solid, and the translational applications are interesting. The English language is fine and does not require revision.

Response:

We thank the Reviewer for the positive evaluation and for the constructive comments. We are confident that, following the Reviewer’s instructions, the manuscript will significantly improve and will be suitable for publication.

I suggest the following improvements of the manuscript:

Comment #1

Line 93 and everywhere else in the text: please use the scientific notation to express big numbers (180,000 should be 1,8 * 10^5)

Response #1

Scientific notation has been employed where appropriate (page 2, line 87; page 3, line 99; page 3, line 126)

Comment #2

Why NAD+ treated group does not show standard deviation in some figures?

Response #2

In Figures 3 and 4 the dynamite plots do not show an error bar for NAD+ treated cells because values are normalized to that condition and therefore, they have no variability.

Comment #3

In the legend of figure 4 there is a mistake “figure 1”. Twice written “of”.

Response #3

The typos were fixed.

Comment #4

Why did authors utilize exactly these cell lines for this study? HAECs are aortic endothelial cells.

Response #4

We thank the Reviewer for raising the insightful comment. We acknowledge that this is a limitation of our in vitro model that is already discussed at page 11, lines 380-385 “Furthermore, the endothelial cells used were of macrovascular origin, rather than microvascular one. This point deserves mentioning since the differences between HAECs and dermal microvascular have not been investigated previously. Nonetheless, HAECs (as well as umbilical vein endothelial cells) are usually considered a reliable universal model to study endothelial biology”.

Comment #5

The quality of images in figure 3 (B and D) should be improved. Inserting a scale bar or specifying magnification would help the readers.

Response #5

We thank the Reviewer for the useful suggestion. We are now providing information about the magnification factor in the Methods (page 4, line 131) and in the caption (page 7, line 232). We are also including a scale bar for the pictures.

Comment #6

In the figures NAD+ is sometimes written without uppercase, in the text with. Please reconcile writing it always with uppercase.

Response #6

We are unsure whether the Reviewer is referring to uppercase or superscript. We did not find any typo for the use of uppercase. We found instead two typos about the superscript (NAD+ instead of NAD+) in Figure 2. The typos were fixed. We hope that we correctly understood the comment and we thank the Reviewer for pointing out these flaws.

Comment #7

Was the liposome matrix tested for toxicity?

Response #7

We thank the Reviewer for this insightful question. We tested the effect of the liposome matrix alone on cell survival and we retrieved the results reported here below (confidential results for the Reviewer).

The liposomal formulation of NAD+ was confirmed as superior to the liposomal matrix alone. Conversely, a significant and unexpected reduction in cell survival was observed in HEKas treated with liposome matrix alone. However, taken together, the inclusion of this additional control does not significantly modify the meaning of results reported in Figure 2, because we still have a significant improvement of cell survival with LIPONAD in HEKas, and a substantially neutral effect in HEKas. Furthermore, the liposomal matrix is not designed to be used alone, rather as an optimal carrier, specifically designed for the chemical properties of NAD+.

Reviewer 2 Report

Comments and Suggestions for Authors

I have reviewed the manuscript (ID: CIMB 3728483) and am not convinced that the authors have developed a novel liposomal delivery system for NAD⁺. While the topic is of interest, there are significant concerns regarding scientific rationale, formulation design, and relevance to the journal's scope.

  1. 1. General Assessment
  • The manuscript is well written, and appropriate literature references are provided.
  • However, the use of the term "novel" in the title is not justified based on the formulation presented. It should be removed.
  • The liposomal components used are well known in literature, and there is no evidence of innovation or significant advancement.
  1. Formulation Concerns

    The authors have utilized the following composition for their liposomal delivery system:

Components

Quantity (%)

Pentylene Glycol

4.95

Lecithin

4.21

Sodium hydroxide

0.32

Tocopherol

0.02

Water

24.10

Glycerin

66.40

   Several critical issues are evident:

  • Use of Sodium Hydroxide (NaOH):
    • NaOH is not commonly used in liposome formation. Its inclusion is neither justified nor explained.
      • NaOH is a strong base that can hydrolyze phospholipids, particularly ester bonds, leading to: (a) Disruption of bilayer integrity; (b) Formation of micelles instead of stable liposomes; (c) Potential degradation of sensitive actives such as NAD⁺
    • Liposomal systems are typically prepared under neutral or slightly acidic conditions to maintain lipid structure and drug stability.
  • Excessive Glycerin Content:
    • The formulation contains an unusually high amount of glycerin (66.4%), which is likely to result in a tacky texture and poor consumer acceptance.
    • Such a formulation lacks commercial viability, and no justification is provided for this choice.
  1. Use of Trade Name
  • The manuscript repeatedly uses the trade name "LIPONAD". This should be avoided in a scientific context. I recommend replacing it with a neutral term such as LFN (Liposomal Formulation of NAD⁺).
  1. Lack of Relevant Data
  • The authors appear to have designed this formulation for topical use but have only presented cell penetration studies.
  • Skin penetration data is essential to validate any topical delivery system and is completely missing from the manuscript.
  1. Suitability for Journal
  • The focus of this work is heavily on formulation development, which does not align well with the journal’s core scientific scope.
  • Unless the authors provide robust mechanistic insights or in vivo efficacy data, this manuscript is more appropriate for formulation or cosmetic science journal.

Recommendation:
I recommend rejection of this manuscript in its current form. The authors must address the significant formulation and scientific shortcomings for the study to be considered credible and of scientific value.

Author Response

Reviewer #2

I have reviewed the manuscript (ID: CIMB 3728483) and am not convinced that the authors have developed a novel liposomal delivery system for NAD⁺. While the topic is of interest, there are significant concerns regarding scientific rationale, formulation design, and relevance to the journal's scope.

Response

We thank the Reviewer for carefully reading our manuscript. We are confident that, thanks to the Reviewer’s constructive comments, we will improve the quality of the manuscript, making it suitable for publication.

Comment #1

  1. 1. General Assessment
  • The manuscript is well written, and appropriate literature references are provided.
  • However, the use of the term "novel" in the title is not justified based on the formulation presented. It should be removed.
  • The liposomal components used are well known in literature, and there is no evidence of innovation or significant advancement.

Response #1

We thank the Reviewer for the insightful comment. The term “novel” was not intended to emphasize a new composition of the liposomal formulation, rather the innovative use of liposomes to encapsulate NAD+. Indeed, at the best of our knowledge there is no other liposomal formulation of NAD+ described in literature or in patents, except for the patent US 12,150,461 B2 (granted 26/11/2024) we reference in the text. However, we recognize that the wording can be misleading, and we have now removed the word “novel” from the title and from the whole manuscript (Abstract, line 26; page 2, line 78; page 10, line 304; page 11, line 363).

  1. Formulation Concerns

    The authors have utilized the following composition for their liposomal delivery system:

Components

Quantity (%)

Pentylene Glycol

4.95

Lecithin

4.21

Sodium hydroxide

0.32

Tocopherol

0.02

Water

24.10

Glycerin

66.40

   Several critical issues are evident:

Comment #2

  • Use of Sodium Hydroxide (NaOH):
    • NaOH is not commonly used in liposome formation. Its inclusion is neither justified nor explained.
      • NaOH is a strong base that can hydrolyze phospholipids, particularly ester bonds, leading to: (a) Disruption of bilayer integrity; (b) Formation of micelles instead of stable liposomes; (c) Potential degradation of sensitive actives such as NAD⁺
    • Liposomal systems are typically prepared under neutral or slightly acidic conditions to maintain lipid structure and drug stability.

Response #2

We thank the Reviewer for the valuable comment. We agree that NaOH is not necessary during the formation of liposomes. Indeed, in the presented formulation, NaOH is only used to adjust the pH after liposome preparation. NaOH is employed as 10% solution and the overall concentration in the liposome solution is <1% (0.32%). In line with the observation of the Reviewer, The NAD+ liposome stock solution is prepared at pH 6.0-6.5. We are now providing additional information about the physical properties of the liposomal formulation over time in Supplementary Table 1, showing that the pH ranges between 6.0-6.5 and is stable for one month. Supplementary Table 1 is referenced in the text at page 2, lines 89-90 “Chemical and physical features of the liposomal matrix and LF-NAD are reported in Supplementary Table 1”.

Comment #3

  • Excessive Glycerin Content:
    • The formulation contains an unusually high amount of glycerin (66.4%), which is likely to result in a tacky texture and poor consumer acceptance.
    • Such a formulation lacks commercial viability, and no justification is provided for this choice.

Response #3

We thank the Reviewer for the practical suggestion. Glycerin was added to the liposomal matrix as a preservative, as it prevents bacterial growth. We include below a table showing that the solution is resistant to bacterial growth for 28 months (confidential results for the Reviewer).

Concerning the possible consumer acceptance, cosmetic products using the described liposomal formulation of NAD+ are already commercially available. The stock solution described in the manuscript is diluted and mixed with other excipients to obtain the final commercial product. However, discussing the commercial viability of specific cosmetic products is, in our opinion, beyond the scope of the paper.

Comment #4

Use of Trade Name

  • The manuscript repeatedly uses the trade name "LIPONAD". This should be avoided in a scientific context. I recommend replacing it with a neutral term such as LFN (Liposomal Formulation of NAD⁺).

Response #4

We agree with the Reviewer that LIPONAD is an unconventional abbreviation for “liposomal formulation of NAD+” that may read like a commercial name. However, it is not a commercial name. To avoid similar misunderstanding, we have now changed the abbreviation of LF-NAD+.

Comment #5

  1. Lack of Relevant Data
  • The authors appear to have designed this formulation for topical use but have only presented cell penetration studies.
  • Skin penetration data is essential to validate any topical delivery system and is completely missing from the manuscript.
  1. Suitability for Journal
  • The focus of this work is heavily on formulation development, which does not align well with the journal’s core scientific scope.
  • Unless the authors provide robust mechanistic insights or in vivo efficacy data, this manuscript is more appropriate for formulation or cosmetic science journal.

Recommendation:
I recommend rejection of this manuscript in its current form. The authors must address the significant formulation and scientific shortcomings for the study to be considered credible and of scientific value.

Response #5

We thank the Reviewer for the valuable suggestion, which has significantly improved the translational significance of the manuscript. Thanks to a collaboration with the University of Paris-Saclay, we now provide additional data about skin penetration ex vivo, showing that the liposomal formulation of NAD+ increases skin absorbance of NAD+ by 30% compared to NAD+ alone. The methods for these additional experiments are now presented on page 4, lines 148-176 “2

2.6. Ex vivo skin penetration

Circular skin explants (Ø= 38 mm) were prepared from the abdominal skin from a Caucasian, 29-year-old, anonymous donor. The collection process fully complied with Articles L.1245-2 and L.1211-1 of the French Public Health Code. The donor provided written informed consent prior to tissue procurement. The explants were prepared as previously described [11]. Briefly, skin samples were held on a specifically designed support composed of a reservoir of culture medium surmounted by a grid on which the skin is stretched. The skin support was connected by a fluidic circuit to a second res-ervoir of culture medium which was stored in the incubator at 37°C, 5% CO2. The circulation of the culture medium was ensured by a peristaltic pump. Each skin explant was treated with 9 µL (2 µL/cm²) of NAD+ 200 μM or LF-NAD+ applied with a small spatula. The control explants did not receive any treatment. After a 24-hour treatment, samples were fixed in buffered formalin solution and included in paraffin for histo-logical sections. 10-µm-thick sections were made using a rotatory microtome (Leica RM 2125, Wetzlar, Germany), and the sections were mounted on a CaF2 specific support for infrared spectroscopy imaging analysis. Based on a preliminary infrared spectral analysis of the pure compounds, NAD+ shows a peak of absorbance at a wavenumber of 1035 cm-1 (Supplementary Figure 2). This peak was used to determine the penetration of NAD+ into the skin. Data collection was performed with a Spotlight 400 FT-IR imaging system (PerkinElmer Inc., Shelton, Connecticut, US), over a spectrum of 750-4000 cm-1. Data were collected over a surface of 0.5 mm2 with a spatial resolution (pixel size) of 6.25*6.25 µm². Hyperspectral images were reconstructed, associating each pixel to a color, representing the intensity of absorbance in that point. Red color indicates a higher intensity, whereas blue indicates a low intensity. The concentration of NAD+ inside of the skin was calculated using the non-negativity constraint classical least squares (NCLS) model. NCLS is a spectral unmixing method that aims to estimate the concentrations of known spectral signatures in a spectrum as previously described [12] to estimate the abundance fractions of a known spectral signature as a linear combination of each component of the signature. The NCLS score is calculated per each pixel and quantified as mean ± SEM”.”.

The corresponding results are reported on page 8, lines 247-255 “According to the above-reported results, LF-NAD+ has an anti-senescence effect on endothelial cells, the main components of the skin microcirculation. Skin microcirculation is organized in two plexuses, parallel to the skin’s surface, to supply oxygen and nutrients to both the outer layer (epidermidis) and the inner layer (dermis) of the skin [12]. So, to exert its beneficial effect on microcirculation, LF-NAD+ should penetrate the skin until the dermo-epidermal junction. Using infrared spectroscopy on human skin explants treated with LF-NAD+ or NAD+ alone, we observed that LF-NAD+ increases the skin penetration of the active substance NAD+ by 30% compared to the application of NAD+ alone (Figure 5).

Finally, the discussion has been adjusted accordingly on page 11, lines 367-370 “This hypothesis is further supported by ex vivo evidence in human skin explants, con-firming the increased bioavailability of NAD+ in skin microcirculation. Additional in-vestigations, such as clinical trials, are needed to confirm that improvements in skin tone and texture can be significantly appreciated by potential users”.

We are confident that, after the extensive revision following Reviewer’s recommendation, the manuscript will be now considered suitable for publication.